# Development, Synthesis and Characterization of Tannin/Bentonite-Derived Biochar for Water and Wastewater Treatment from Methylene Blue

**Mariia Galaburda** [1,2] , **Alicja Bosacka** [2,*] , **Dariusz Sternik** [2], **Viktor Bogatyrov** [1], **Olena Oranska** [1], **Volodymyr Gun'ko** [1] **and Anna Deryło-Marczewska** [2]

[1] Chuiko Institute of Surface Chemistry, NAS of Ukraine, 17, General Naumov Street, 03164 Kyiv, Ukraine; mariia.galaburda@gmail.com (M.G.); vbogatyrov@gmail.com (V.B.); el.oranska@gmail.com (O.O.); vlad_gunko@ukr.net (V.G.)

[2] Institute of Chemical Sciences, Faculty of Chemistry, Maria Curie-Skłodowska University, 3, Maria Curie-Sklodowska Square, 20-031 Lublin, Poland; dariusz.sternik@mail.umcs.pl (D.S.); anna.derylo-marczewska@mail.umcs.pl (A.D.-M.)

[*] Correspondence: alicja.bosacka@poczta.umcs.lublin.pl

**Abstract:** Novel hybrid carbon–mineral materials were synthesized by the mechanochemical activation of a mixture of tannin and bentonite in a ball mill with further pyrolysis in an argon atmosphere at 800 °C. The influence of the initial mixture ingredients content on the structural, textural, and thermal characteristics of biochars has been described using X-ray diffraction, Raman and Fourier-transform infrared spectroscopy, nitrogen adsorption–desorption, and scanning electron microscopy. The influence of bentonite clay on the carbon phase characteristics due to the formation of more heat-resistant and structured nanocarbon particles in biochars has been proven. The adsorption effectiveness of the materials towards methylene blue was studied. The adsorption data were analyzed applying Langmuir and Freundlich isotherms with high determination coefficients ($R^2$) in the range of 0.983–0.999 (Langmuir) and 0.783–0.957 (Freundlich). The maximum adsorption amount of MB was 5.78 mg/g. The adsorption efficiency of biochars with respect to phenol was also examined. It was shown that the hybrid biochars show differentiated selectivity to the adsorption of organic compounds. It was concluded that the physicochemical properties of the surface of biochars play an important role in the adsorption effectiveness, making them a good candidate for water and wastewater remediation processes.

**Keywords:** tannin; bentonite; biochar; composite; pyrolysis; dye adsorption

## 1. Introduction

Dyes are substances of natural or synthetic origin that give color to other substances. Synthetic dyes, which are compounds with a complex aromatic structure of molecules, are especially hazardous and hard to be decomposed in water. One of the synthetic dyes is methylene blue (MB), also known as basic blue 9—whose chemical name is methylthioninium chloride—which is characterized by the basic character, the molar mass of 319.9 g/mol, the water solubility of 4.36 g/100 mL at 25 °C, and ionization constant (pKa) of 3.8. Methylene blue in the solid state is green crystalline powder, but dissolved in water is blue. Methylene blue is used in the textile industry, however, is considered as a toxic, carcinogenic, non-biodegradable, and environmentally dangerous substance, therefore, its removal from technological wastewater and water is of great importance [1–3]. There are many methods of purifying water and wastewater from dyes. The adsorption is one of the leading routes in terms of use simplicity, adsorption efficiency, and great possibilities in adsorbent design [4,5]. There are many various appropriate adsorbents. However, new, selective materials with enhanced adsorption effectiveness are still sought. In this work,

the novel hybrid carbonized materials were prepared by the mechanochemical activation of ball-milled and subsequently carbonized blends of tannin and bentonite to be applied for the adsorption of methylene blue.

An initial mixture was prepared by the mechanochemical activation. It should be noted that mechanochemistry is a way of material receiving in which the chemical reaction's activation energy is supported by a mechanical force [6]. The chemical changes of solid-state starting materials are initialized by high-energy mills or grinders [7,8]. Additionally, mechanochemical activation using mills or grinders is one of the key methods for the synthesis of nanostructured materials [9,10].

One of the ingredients of the used blends is tannin, which is a natural, versatile, non-toxic aromatic polyphenolic compound. Tannin is an amorphous light-yellow powder with a slightly peculiar odor. Tannins are present in almost all parts of plants such as bark, wood, leaves, fruits, roots and seeds. Soluble tannin (tannic acid) consists of units with glucose linked by ester bonds with nine to ten molecules of gallic acid. The presence of poorly water-soluble polyphenols (phloroglucinol and resorcinol) in tannins or the formation of condensed tannins due to condensation of hydroxyls from neighboring molecules can reduce their solubility in water [11]. Tannins are one of the most common biomass compounds similar to cellulose, hemicelluloses, and lignin [11]. Drugs for internal and external use can be made using tannins immobilized on a surface of nanosilica [12]. Tannins can be used in many industrial domains: food [13], ecology [14], and medicine in the form of technical and pharmaceutical compounds [15,16]. They possess antimicrobial and immunomodulatory properties [17]. As a main raw material in the present work, grape seeds were used, separated from the unfermented pulp during the wine making season. In grapes, tannins are found in the skins, as well as in the seeds and stems. Grape and wine tannins are condensed tannins produced by the polymerization of several plant-derived flavonoid molecules [18]. Thus, given the above, the replacement of phenolic materials (i.e., phenol and resorcinol) with tannin for the synthesis of carbon sorbents can be beneficial to both the bioeconomy and environment due to its low-cost, bio-origin, non-toxic and non-carcinogenic properties.

The second component of the reaction mixture is bentonite, a natural origin clay that is formed from volcanic ash. Bentonite consists mainly of crystalline clay minerals belonging to the smectite group, which are hydrous aluminium silicates containing a mixture of sodium, magnesium, calcium, and iron. Structurally, a 2:1 layer of bentonite consists of two silica tetrahedral sheets with a central octahedral alumina sheet. The space between these layers is occupied by structural water and cations. The bentonite layers are negatively charged and held together by charge-balancing counterions such as $Na^+$ and $Ca^{2+}$. Water and counterions in the 2:1 layer in bentonite separate the clay layers [19]. Such a structure of the mineral makes it possible to modify it by filling interlayer spaces while maintaining the layered structure (2:1 layer) and increasing the specific surface area, pore volume, and amount of active centers on the surface of the material. Typically, montmorillonite is the main component of bentonite and consists of mainly $SiO_2$, $Al_2O_3$ with $Fe_2O_3$, MgO, $Na_2O$, and $K_2O$, which can promote the carbon yield [20]. Bentonite incorporated into biomass increases the pyrolysis gas yield, which can act as additional gas for activation during pyrolysis and carbon formation [21]. Bentonite can be used as a catalyst in pyrolysis, cracking, and biodiesel processes that may enhance their efficiency [21–23]. In a previous study, it was demonstrated how the way of incorporation of synthetic resorcinol–formaldehyde resins into bentonite clay can influence the porosity and activity of hybrid adsorbents [24]. Biochars produced at high temperatures (>700°C) are more effective for organic contaminant sorption because of their strong carbon structurization, relatively high specific surface area, and porosity [25]. Biochar surfaces become less polar and more aromatic after a high-temperature carbonization, which may affect the organic contaminant adsorption [26,27].

The starting mixture of tannin and bentonite was pyrolyzed at a high temperature and an inert gas atmosphere to produce the biochars. It is worth mentioning that the

carbonaceous materials are considered to be effective sorption systems for the removal of organic pollutants from water and wastewater because of their well-developed surface area and porosity. The biochars could be produced due to the thermochemical decomposition of agricultural residues under restricted oxygen conditions. In the process of carbonaceous materials production, the initial components are subjected to pyrolysis and activation which are extremely important preparation steps in the formation of porous materials with appropriate chemical properties increasing adsorption efficiency [28–30]. Furthermore, pyrolysis of certain blends allows for the synthesis of novel hybrid materials retaining the structural advantages of bentonite (mesoporosity) and adding new structural characteristics due to a carbon component.

In the presented paper, the structural, textural, morphological, and thermal characteristics of biochars prepared by mechanochemical activation and pyrolysis at 800 °C in an inert gas atmosphere of blends of tannin and bentonite of differentiated compositions have been investigated by several techniques: low-temperature nitrogen sorption, thermogravimetry (TG), X-ray diffraction (XRD), Fourier transform-infrared (FT-IR) and Raman spectroscopies, and scanning electron microscopy (SEM). The adsorption equilibrium studies of the carbonized materials towards methylene blue were carried out. The adsorption data were analyzed by applying Langmuir and Freundlich isotherms. The phenol adsorption studies were also performed for a better understanding of the sorption mechanism by biochars. The adsorption studies revealed the applicability of the obtained adsorbents in water and wastewater treatments to remove the organic substances of various properties.

## 2. Materials and Methods

### 2.1. Materials and Chemicals

The experiments were conducted using bentonite (Sigma-Aldrich, Gillingham, UK). Tannin products (soluble in alcohol, extracted from grapes purchased from a local wine merchant in the Republic of Moldova. The hydrochloric acid (35–38%), sodium hydroxide, and sodium chloride were bought from Chempur (Piekary Slaskie, Poland). The methylene blue (MB) was purchased from Merck (Darmstadt, Germany). The solutions for adsorption studies were prepared with the use of deionized water (deionizer Polwater, Labpol, Cracow, Poland).

### 2.2. Materials Preparation

A two-stage method of synthesis was used: (i) mechanochemical activation of a mixture and (ii) pyrolysis. In the first part, the certain weights of tannin and bentonite were placed into a stainless steel jar (250 mL) containing stainless steel balls (5 mm in diameter) and milled for 6 h in a laboratory ball mill. The second step was pyrolysis in the quartz cells of the vertical reactor in an argon flow (100 mL/min) at 800 °C. The heating rate was 10 °C/min and a sample was maintained for 2 h at the set temperature. Cooling was carried out to room temperature in an argon flow. The names of the samples and the initial ratios of the components are given in Table 1.

**Table 1.** The ratios of the initial components used to prepare materials.

| Sample | Component Ratio, g | |
|---|---|---|
| | Tannin | Bentonite |
| TC (control) | 10 | - |
| TBC-1 | 10 | 10 |
| TBC-2 | 20 | 10 |
| TBC-3 | 30 | 10 |
| TBC-0.5 | 6 | 10 |

### 2.3. Methods

X-ray diffraction (XRD) patterns were recorded at 2θ = 5–80° using a DRON-4-07 diffractometer with filtered Cu Kα radiation in the geometry of Bragg–Brentano. The phases were identified using the PDF-2 X-ray database of standards.

The Raman spectra were recorded over the 3200–150 cm$^{-1}$ range using a Via Reflex Microscope DMLM Leica Research Grade, Reflex (Renishaw, Gloucestershire, UK) with laser excitation at λ$_0$ = 785 nm.

The FT-IR spectra were registered in the range of 400–4000 cm$^{-1}$ using Tensor 27 apparatus (Bruker, Selb, Germany) with an ATR attachment equipped with a diamond crystal.

A morphological analysis of the studied materials was carried out using a Quanta 250 FEG scanning electron microscope (FEI, Hillsboro, OR, USA) equipped with an EDS attachment.

To compute the textural characteristics, the low-temperature (77 K) nitrogen adsorption–desorption isotherms were recorded using an ASAP 2405N (Micromeritics, Norcross, GA, USA) adsorption analyzer. The specific surface area, $S_{BET}$, was calculated according to the standard BET method [31]. The total pore volume, $V_p$, was evaluated from the nitrogen adsorption at p/p$_0$ = 0.98–0.99 (p and p$_0$ denote the equilibrium and saturation pressure of nitrogen at 77.4 K, respectively). The nitrogen desorption data were used to compute the pore size distributions (PSD, differential $f_V(R) \sim dV_p/dR$ and $f_S(R) \sim dS/dR$), using a model with slit-shaped pores for carbon–bentonites [32]. The differential PSD concerning pore volume, $f_V(R) \sim dV_p/dR$ and $\int fV(R)dR \sim V_p$, were recalculated as incremental PSD (IPSD), $\sum \Phi_{v,i}(R) = V_p$. The $f_V(R)$ and $f_S(R)$ functions were also used to calculate the contributions of micropores ($V_{micro}$ and $S_{micro}$ at radius R ≤ 1 nm), mesopores ($V_{meso}$ and $S_{meso}$ at 1 nm < R < 25 nm) and macropores ($V_{macro}$ and $S_{macro}$ at 25 nm < R < 100 nm) to the total pore volume and specific surface area.

A potentiometric titration of suspension was carried out using a multicomponent system equipped with a pH meter (PHM240 Radiometer, Copenhagen, Denmark), an autoburet 765 Dosimat (Metrohm, Herisau, Switzerland), a thermostat (Ecoline RE207, Lauda, Germany), a degassing system (nitrogen atmosphere), and a quartz vessel. In the first step was 30 mL of 0.1 mol/dm$^3$ NaCl (an electrolyte) with the addition of 300 μL of 0.2 HCl, the solution then equilibrated for 1 h at 25 °C to achieve constant temperature and pH value, then it was titrated with 0.1 mol/dm$^3$ NaOH. In the second step, the same procedure was repeated but with the addition of 1 g of solid sample. Finally, the surface charge density and the point of zero charge (pH$_{PZC}$) were determined.

The thermal stability of samples, as well as the carbon content, was determined using an STA 449 Jupiter F1 (Netzsch, Selb, Germany) apparatus. The samples (~12 mg) were heated at a rate of 10 °C/min in the range of 30–950 °C in the atmosphere of synthetic air (flow of 50 mL/min) in an alumina crucible, and sensor thermocouple type S TG–DSC. An empty Al$_2$O$_3$ crucible was used as a reference. Thermogravimetry (TG and DTG) curves were registered during the analysis.

Adsorption equilibrium studies were performed applying a Carry 4000 UV-Vis spectrophotometer (Varian, Melbourne, Australia). Initially, the 50 mL conical flask with an adsorbent of mass ~0.05 g and 25 mL solution of MB with the desired concentration in the range of 1.25–40 mg/L and pH~7was placed into the incubator shaker for 2 days at 25 °C (110 rpm) (New Brunswick Scientific, Edison, NJ, USA). After the equilibrium achievement, the absorbance was measured at 200–800 nm, and the organic substance concentration was determined from the maximum absorbance peak A = f(λ).

The adsorbed amount (*a*) was calculated based on the mass balance equation:

$$a = \frac{(C_0 - C_{eq}) \cdot V}{m} \tag{1}$$

where: *a* is the adsorption amount (mg/g), $C_0$—the initial concentration of dye (mg L$^{-1}$), $C_{eq}$—the equilibrium concentration of dye (mg L$^{-1}$), *V*—the volume of solution, and *m*—the mass of adsorbent.

The analysis of the adsorption process was carried out based on the Langmuir (1) and Freundlich (2) isotherms using the equations:

$$\frac{C_{eq}}{a} = \frac{C_{eq}}{a_m} + \frac{1}{K_L \cdot a_m} \tag{2}$$

$$\ln a = \ln K_F + \frac{1}{n} \cdot \ln C_{eq} \tag{3}$$

where: $a_m$ is the adsorption capacity (mg/g), $C_{eq}$—the equilibrium dye concentration (mg L$^{-1}$), $K_L$—the Langmuir constant related to the rate of adsorption (L/mg), $K_F$—the Freundlich constant (mg/g), and $n$—the parameter describing the favorableness of the sorption process.

To compare, the additional adsorption studies towards phenol were carried out using a similar procedure. The adsorbent of mass 0.02 g was placed in 50 mL Erlenmeyer flasks with 10 mL of phenol solution ($C_0$ = 1.054 mmol/L) and maintained for 24 h in an incubator shaker (New Brunswick Scientific, Edison, NJ, USA, 110 rpm, 25 °C). Then carbon slurry was centrifugated (MPM 252R, 10,000 rpm for 10 min). The absorbance was measured at 200–450 nm, and the phenol concentration was determined from the maximum absorbance peak measured using the Carry 4000 UV-Vis spectrometer ($\lambda$ = 270 nm).

The effect of pH on the percent of removal of MB was analyzed using TBC-0.5 sample. The mixtures were prepared in plastic containers (10 mL) by addition of the methylene blue solution with a defined pH (5 mL of MB with an initial concentration of 10 mg/L) to 0.01 g of adsorbent. The adsorption in the examined system was carried out at pH: 3, 5, 7, 9, and 11 (corrected by the addition of 0.1 mol/dm$^3$ HCl or 0.1 mol/dm$^3$ NaOH solution). The solution was stirred using a shaker (New Brunswick Scientific, Edison, NJ, USA, 110 rpm, 25 °C). When equilibrium was attained, the adsorbent was separated bycentrifugation. Then, the supernatant was analyzed in terms of residual concentration.

The influence of contact time on the MB removal (%) was also investigated taking into account the TBC-0.5. The suspensions were prepared in plastic containers (10 mL) by the addition of the MB solution (5 mL, $C_0$ = 20 mg/L) to 0.01 g of adsorbent. The adsorption was carried out for selected times: 1, 3, 6, 12, 24, and 48 h. When equilibrium was achieved, the adsorbent was separated by centrifugation. Analogous to the pH studies, the residual concentration was determined.

## 3. Results

### 3.1. Structural Characteristics

XRD patterns of the samples are given in Figure 1. The characteristic peaks on the pattern of the natural bentonite indicate that the clay is composed primarily of montmorillonite with some impurities corresponding to quartz and muscovite (JCPDS No. 3-19, 85-795, 7-42, respectively, Figure 1a, Table 2). After pyrolysis at 800 °C a partial destruction of the layered and crystalline structure of montmorillonite took place that is accompanied by the disappearance of the basal peak (001) at 2θ = 6° (interplanar distance 1.47 nm) and other peaks. The patterns of carbonized samples indicate that the structure of bentonite remains the same as in treated bentonite alone. The raise in tannin content lead to an increase in the content of carbon during carbonization. An increase in the intensity of the diffuse halo at 2θ = 25° and a decrease in the intensity of the quartz and muscovite peaks indicate an increase in the content of amorphous carbon in the biochar. Considering the carbonized raw tannin, there are two typical diffuse halos at 2θ = 25° and 50°, characteristic of amorphous carbon, and are well recorded in the spectrum. Thus, the synthesized materials are a combination of particles of bentonite with a destructed structure and amorphous carbon, which is partially formed in the interplanar space of bentonite layers.

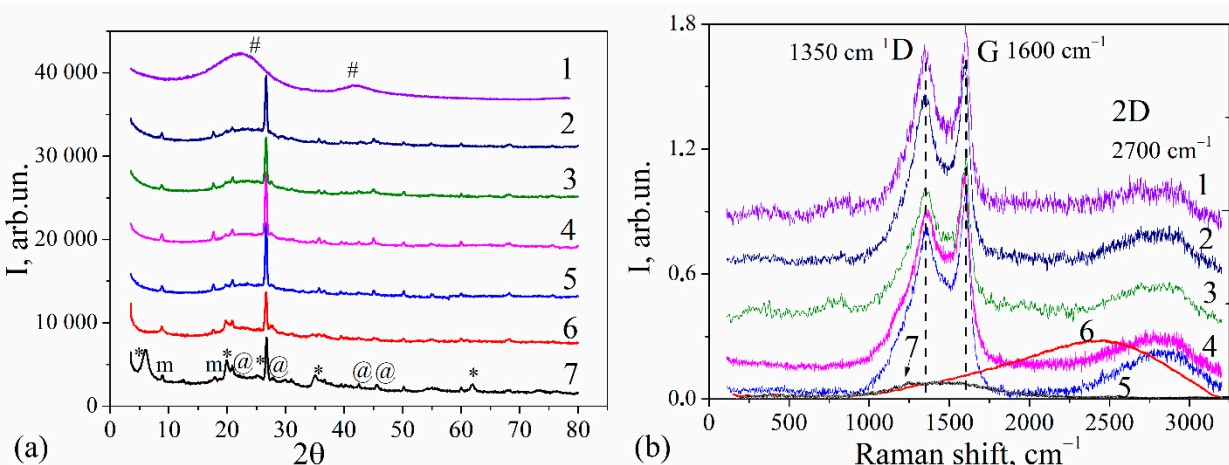

**Figure 1.** X-ray diffraction patterns (**a**) and Raman spectra of the pyrolyzed materials (**b**): 1—TC (control), 2—TBC-3, 3—TBC-2, 4—TBC-1, 5—TBC-0.5, 6—bentonite pyrolyzed, 7—bentonite (*—montmorillonite, @ quartz, m—muscovite, and #—amorphous carbon).

**Table 2.** Structural characteristics of pyrolyzed tannin–bentonite materials.

| No. | Sample | PDF-2 Card No. | Phase | FWHM ($cm^{-1}$) | | Center ($cm^{-1}$) | | D/G Intensity Ratio ($I_D/I_G$) |
| --- | --- | --- | --- | --- | --- | --- | --- | --- |
| | | | | D-Band | G-Band | D-Band | G-Band | |
| 1 | TC (control) | - | Amorphous carbon | 126.3 | 126.3 | 1340.4 | 1598.0 | 0.8 |
| 2 | TBC-1 | -<br>85–795<br>7–42<br>- | Montmorillonite$_{destr.}$<br>Quartz<br>Muscovite<br>Amorphous carbon | 145.3 | 145.3 | 1354.8 | 1590.0 | 0.8 |
| 3 | TBC-2 | -<br>85–795<br>7–42<br>- | Montmorillonite$_{destr.}$<br>Quartz<br>Muscovite<br>Amorphous carbon | 155.6 | 154.8 | 1349.6 | 1600.2 | 0.8 |
| 4 | TBC-3 | -<br>85–795<br>7–42<br>- | Montmorillonite$_{destr.}$<br>Quartz<br>Muscovite<br>Amorphous carbon | 144.0 | 144.0 | 1340.8 | 1600.2 | 0.8 |
| 5 | TBC-0.5 | -<br>85–795<br>7–42<br>- | Montmorillonite$_{destr.}$<br>Quartz<br>Muscovite<br>Amorphous carbon | 145.3 | 145.3 | 1352.9 | 1600.2 | 0.9 |
| 6 | Bentonite | 3–19<br>85–795<br>7–42 | Montmorillonite<br>Quartz<br>Muscovite | - | - | - | - | - |
| 7 | Bentonite (pyrolyzed) | -<br>85–795<br>7–42 | Montmorillonite$_{destr.}$<br>Quartz<br>Muscovite | - | - | - | - | - |

Raman spectroscopy was used to obtain additional information about the carbon phase in biochars. The spectra were collected in the most informative range for carbon materials of 200–3200 $cm^{-1}$ (Figure 1b). The spectra of biochar were characterized by two distinctive bands, G (at ~1600 $cm^{-1}$), and D (at ~1350 $cm^{-1}$) [33]. The G-band represents in-plane vibrations of $sp^2$ hybridized carbon atoms in the graphitic structure, and the D-band is derived from the partial carbon lattice defects that can be mainly due to the structural distortion (breathing mode of aromatic rings) [34]. During graphite amorphisation, both

bands changed. The broad G, as well as D bands of high relative intensity, indicate high disordering of carbon. The ratio of the intensities of the D- and G-bands ($I_D/I_G$) is often used as an indicator of graphitization degree, i.e., the contribution of plane polyaromatic structures in the materials. As can be seen in Table 2 the materials were graphitized with defects and disorders. The $I_D/I_G$ value did not show significant change and thus, one can state that bentonite does not affect the structural features of the carbon phase.

Moreover, a series of peaks in the region of 2500–3000 cm$^{-1}$ could be detected. In particular, the broad 2D peak at ~2700 cm$^{-1}$, which is called the overtone mode of the D mode, indicates a highly amorphous skeleton containing graphitic domains in biochar of a few nanometres in size (less than 5 nm due to the XRD data). The higher the intensity of this bend, the smaller the size of the carbon crystalline. Thus, the size of graphitic component increased with the following tendency: TBC-0.5 < TBC-1 < TBC-2 < TBC-3 < TC (control).

The spectrum of raw bentonite corresponds to the spectrum of montmorillonite (the main mineral in bentonite) according to the Raman spectroscopy databases, which includes the stretching vibrations of hydroxyls and water (ν3 vibration of $H_2O$, ν3(Me–$H_2O$) [35]. After carbonization, the overlapped and not so intensive bands could be observed in the region of 900–2000 cm$^{-1}$, which correspond to ν3($SiO_4$) and ν1($SiO_4$) vibrations. The weak bands in the spectrum at 580 and 850 cm$^{-1}$ could be referred to Al–O and Fe–O fragments.

The FT-IR spectra of bentonite and tannin–bentonite biochars are shown in Figure 2. The intensive and sharp band at 3600 cm$^{-1}$ and broad band at 3400 cm$^{-1}$ belonged to OH stretching vibrations (ν3) of structural hydroxyl groups and water present in the raw bentonite due to hydroxyl bonding between octahedral and tetrahedral layers in the mineral [36]. Another sharp peak at 1630 cm$^{-1}$ was assigned to the asymmetric OH of water (deformation vibrations of bound water). The intensive bands centered at ~1000 cm$^{-1}$ and ~1020 cm$^{-1}$ were assigned to the silicate montmorillonite mineral and attributed to the Si-O stretching vibrations ν3 (in-plane) [34]. The OH deformation mode of Al-O-Al-OH or Al-OH-Al was recorded at 914 cm$^{-1}$ [37]. The band centered at 690 cm$^{-1}$ belonged to the deformation and bending modes of the Si–O bonds, while at 548 cm$^{-1}$, to the deformation mode of the Al-O-Si group. Several bands in the lower region were due to the vibrational modes of the $SiO_4$ units [34].

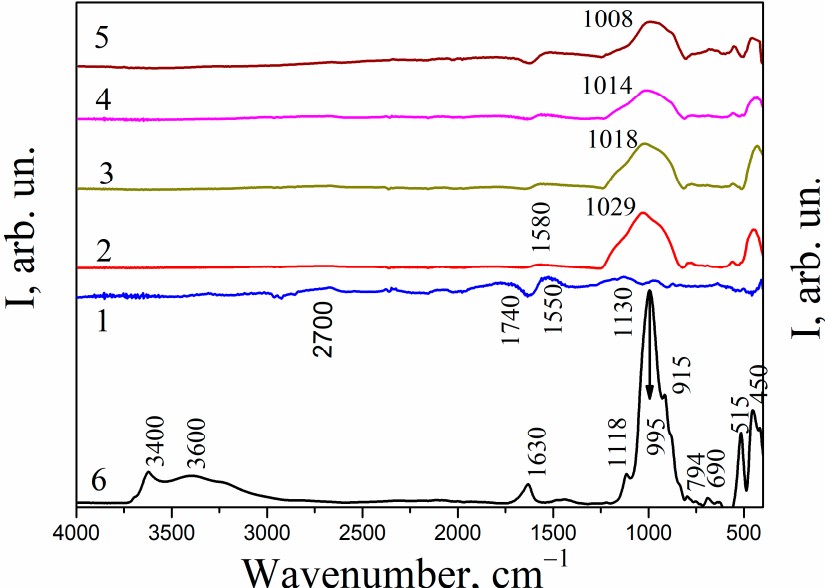

**Figure 2.** FT-IR spectra of carbonized tannin–bentonite materials: 1—TC (control), 2—TBC-3, 3—TBC-2, 4—TBC-1, 5—TBC-0.5, 6—bentonite.

In the biochar, significant changes in the surface structure could be seen in the IR-spectra due to the dehydration and decarboxylation processes associated with the formation of

structures containing aromatic carbon. The characteristic C–H stretching vibrations of the alkyl structure of the aliphatic group (C–H stretching) in the TC (control) sample could be seen in the region of 2580–2950 cm$^{-1}$. In addition, the presence of a skeletal vibration spectrum, typical for aromatic lignin fragments (C=C and –OCH$_3$) could be observed in the region of 1510–1450 cm$^{-1}$ [38]. A band at 1580 cm$^{-1}$ was related to the C=C aromatic ring stretching vibrations. The band occurring in the region of 1500–1600 cm$^{-1}$ could be attributed to the C=C stretching vibrations of aromatic components and to a smaller extent to the C=O stretching vibrations in quinones and ketonic acids [38], and the aromatic C–C and C–O stretching vibrations of conjugated ketones (quinones). A band at 1580 cm$^{-1}$ belonged to COO– asymmetric stretching vibrations. The wide band at 1740 cm$^{-1}$ could be attributed to the C=O stretching vibrations which indicates the presence of the carboxylic group in the surface of biochar [39]. A band at 1550 cm$^{-1}$ could be referred to the COO– asymmetric stretching vibrations while the band occurring at 1130 cm$^{-1}$ was characteristic to the C–O–C stretching vibrations [40]. Analyzing the spectra of mineral-containing biochars, it can be seen that the surface of the TBC-0.5 sample (Figure 2, curve 5) had more intensive bands in the region of 1500–1800 cm$^{-1}$ due to the greater aromatic character of the carbon component and a high degree of carbonization. The decrease in peak intensity in the lower region with almost complete disappearance at 3000–3700 cm$^{-1}$, as well as bands shift indicated significant changes and damages of the layered structure of bentonite (dihydroxylation) in the biochar after carbonization at 800 °C. However, a significant broadening of a peak at ~1000 cm$^{-1}$ in the pyrolyzed materials, with simultaneous shift to the red region (1028–1008 cm$^{-1}$) may indicate the formation of chemical bonds, such as in the Si–O–C groups [39–41].

### 3.2. Thermal Properties

The TG curves for the carbonized materials (Figure 3) present a major mass loss step in the temperature range of 350–650 °C with a corresponding effect on the DTG curves. This mass loss was related to the decomposition of the carbon skeleton with the release of CO$_2$, while the final residue corresponds to the dehydrated bentonite. The content of carbon in the samples after carbonization changed in proportion to the amount of tannin in the initial blends. The amount of physically adsorbed water was less than 1 wt% (desorbed up to 300 °C), so it was not taken into account in the calculations. Bentonite content enhanced the temperature of the thermo-oxidative destruction of carbon. The control sample (TC) had the lowest T$_{max}$ = 550 °C, and in the case of other samples with different carbon content in the material, a slight differentiation of T$_{max}$ was observed.

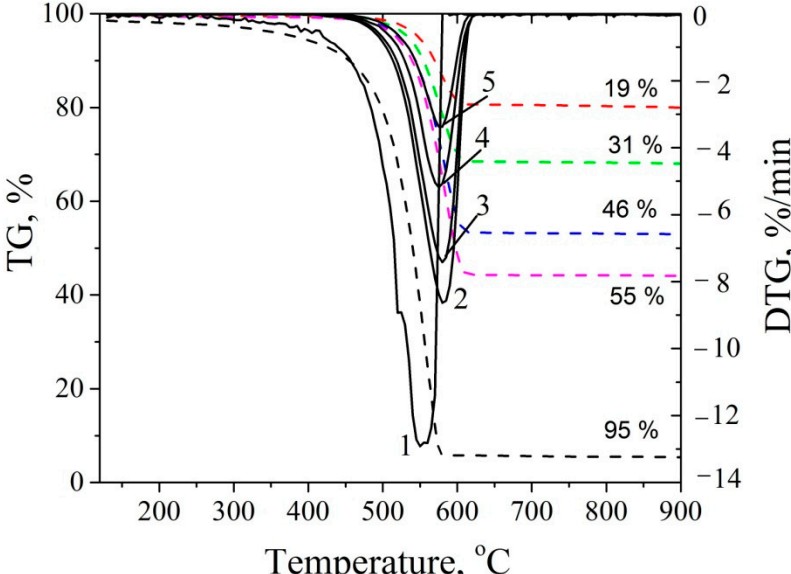

**Figure 3.** TG and DTG curves of biochars: 1—TC (control), 2—TBC-3, 3—TBC-2, 4—TBC-1, 5—TBC-0.5.

It can be noted that with increasing bentonite content in the material, an increase in the oxidation temperature was observed from 577 °C (TBC-3 sample) to 585 °C (TBC-0.5). Thus, the enhancement of thermal stability of biochar took place due to the creation of Si-O-C bonds in the samples and location of carbon layers between clay layers.

### 3.3. Morphological Properties

Scanning electron microscopy (SEM) was used to evaluate the morphology of the carbonized materials. The SEM images (Figure 4) of the pyrolyzed samples show that carbon is in the form of irregularly shaped particles of 1–1.5 μm sizes and consists of X-ray amorphous carbon structures (Figure 1), formed on lamellar polygonal bentonite flakes. With an increase in the content of tannin (a carbon source), the size of particles practically did not change, but their density increased. Moreover, the original structure of the clay mineral was broken after carbonization, and the polygonal lamellar morphology was disturbed with decreasing size and shape of individual plate particles. As a result of the introduction of carbon into the interplanar spacing of clay during modification processes and pyrolysis, the structural degradation of bentonite was observed. The residues become much looser, and the pore sizes decreased. This statement was checked further based on nitrogen sorption studies.

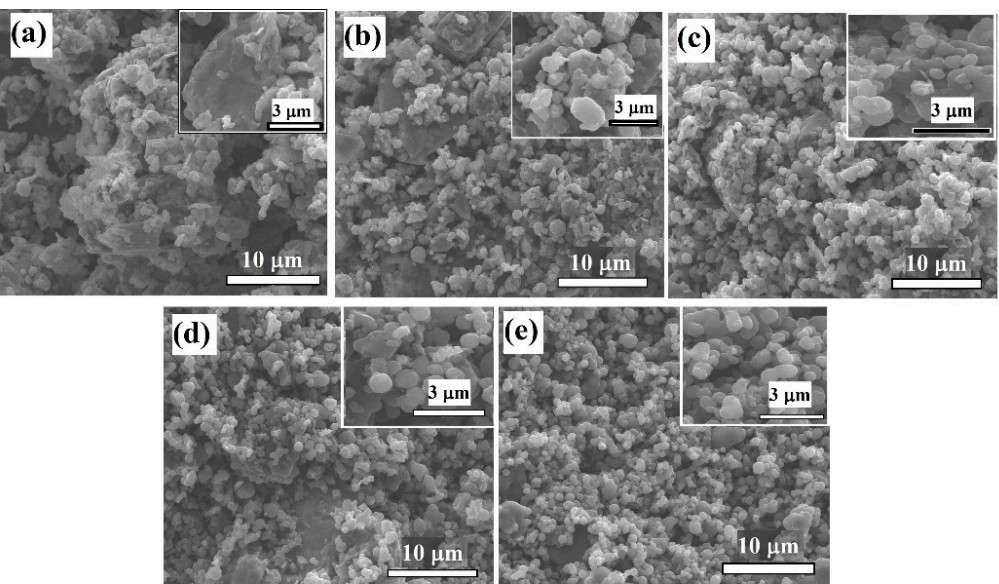

**Figure 4.** SEM images of TBC-0.5 (**a**), TBC-1 (**b**), TBC-2 (**c**), TBC-3 (**d**), and TC (control) (**e**).

### 3.4. Porosity and Acid–Base Properties

Nitrogen sorption isotherms of pyrolyzed tannin–bentonite materials can be classified as a mixture of Type I and IV isotherms (IUPAC) and are characterized by rapidly increased adsorption at low relative pressure ($p/p_0$) and slowing down at higher relative pressures. Thus, monolayer adsorption at low relative pressure, multilayer adsorption at a higher relative pressure, and capillary condensation in mesopores (Figure 5a). This hysteresis loop may be classified as a type H4, which can be associated with narrow slit-like pore structures and its elongation to lower relative pressures, indicating the presence of micropores in the material. The raw bentonite shows Type IV adsorption isotherms with a large uptake near saturation pressure, indicating the mesopores structure.

The BET surface area of the investigated materials and porosity data are summarized in Table 3. It is worth noting that the layered structure of bentonite collapsed after pyrolysis due to the removal of interlamellar water and dihydroxylation, and as a result, it became less porous. Compared to raw bentonite (Table 3), the carbonized biochars are characterized by a higher BET surface area and the contribution of micropores to the total porosity of

the materials (Figure 5b). It was shown that an increase in the tannin content raised the volume of nanopores due to the formation of the carbon phase during the pyrolysis process occurring in the interlayer space of bentonite.

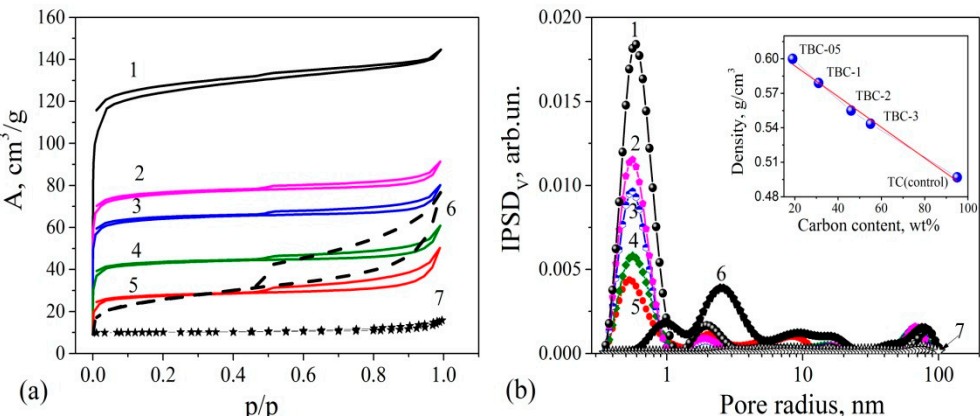

**Figure 5.** Nitrogen adsorption/desorption isotherms (**a**) and pore size distributions (**b**) of biochars: 1—TC (control), 2—TBC-3, 3—TBC-2, 4—TBC-1, 5—TBC-0.5, 6—bentonite, 7—bentonite pyrolyzed. Bulk density versus carbon content is in the insertion.

**Table 3.** Porous characteristics and $pH_{PZC}$ for the studied materials.

| No. | Sample | $S_{BET}$ (m²/g) | $S_{micro}$, m²/g | $S_{meso}$, m²/g | $S_{macro}$, m²/g | $V_p$ (cm³/g) | $V_{micro}$ (cm³/g) | $V_{meso}$ (cm³/g) | $V_{macro}$ (cm³/g) | $V_{micro}/V_p$ | $V_{meso}/V_p$ | $pH_{PZC}$ |
|---|---|---|---|---|---|---|---|---|---|---|---|---|
| 1 | TBC-0.5 | 86 | 75.2 | 9.0 | - | 0.08 | 0.05 | 0.03 | 0.00 | 0.61 | 0.39 | 8.55 |
| 2 | TBC-1 | 128 | 121.1 | 6.2 | 0.2 | 0.09 | 0.06 | 0.02 | 0.01 | 0.72 | 0.20 | 6.41 |
| 3 | TBC-2 | 194 | 188.2 | 5.4 | 0.2 | 0.13 | 0.10 | 0.02 | 0.01 | 0.80 | 0.13 | 6.39 |
| 4 | TBC-3 | 230 | 223.4 | 6.2 | 0.2 | 0.14 | 0.11 | 0.02 | 0.01 | 0.81 | 0.12 | 5.84 |
| 5 | TC(control) | 380 | 368.5 | 10.8 | 0.1 | 0.23 | 0.20 | 0.02 | 0.01 | 0.86 | 0.10 | 6.95 |
| 6 | Bentonite | 86 | 28.3 | 57.0 | 0.6 | 0.12 | 0.01 | 0.08 | 0.02 | 0.08 | 0.70 | 7.90 |
| 7 | Bentonite pyrolyzed | 3 | 0.6 | 1.9 | 0.2 | 0.01 | - | 0.01 | 0.00 | 0.59 | 0.39 | 8.42 |

With an increase in carbon content, an increase in the hydrophobicity of biochar took place, which lead to a decrease in their wetting by water. The TBC-0.5 sample is characterized by the highest wettability of the series of pyrolyzed materials due to a smaller amount of carbon, carbonization, and a lower degree of graphitization ($I_D/I_G = 0.9$). Higher hydrophilicity can be due to a small content of hydrophobic aromatic carbon structures (19%) and a significant contribution of bentonite (81%).

The bulk density of a particulate filler depends on the size, shape, surface structure, and particle packing. A change in the bulk density in the samples is directly proportional to the carbon content since the carbon phase has bulk and true densities smaller than that of clay. This fact confirms that the porosity of the samples is affected by the filling with carbon in the process of pyrolysis of tannin (Figure 5b, insertion).

The acid–base properties of tannin–bentonite biochars were investigated by potentiometric titration of the aqueous suspension. Generally, the values of point of zero charge ($pH_{PZC}$) for investigated samples are given in Table 3 and are in the range of 5.84–8.55. The changes in the $pH_{PZC}$ values are associated with the different composition of the initial mixtures. For the TC (control) sample, the $pH_{PZC}$ value is ~7 which identifies an inert character of its surface. During the pyrolysis of the TC sample, the most surface functional groups (e.g., phenolic) were removed. For TBC-3, TBC-2, and TBC-1, the $pH_{PZC}$ values are 5.84, 6.39, and 6.41 that are related to the existence of certain oxygen species of acidic character. The TBC-0.5, bentonite, and pyrolyzed bentonite are characterized by higher $pH_{PZC}$ values (8.55, 7.90, and 8.42) that reveal the presence of some amount of basic centers on their surfaces [41].

### 3.5. Adsorption Characteristics

The scheme of synthesis of biochars from tannin and bentonite with possible mechanisms for methylene blue and phenol adsorption is presented in Figure 6. Figure 7a presents the isotherms of methylene blue (MB) adsorption on biochars. The effect of pH and contact time on MB removal by the adsorbent is shown in Figure 7b. The results of phenol removal by TBC-0.5, TBC-1, TBC-2, TBC-3 and bentonite are shown in Figure 8. The effect of contact time and pH on phenol removal by TBC-0.5 is shown in Figure 9. The studies showed a significant increase in the adsorption capacity of the materials prepared with the addition of aluminosilicate in the reaction blends in comparison to the control sample (TC). It may be correlated with the highest share of micropores in TC porosity (0.86) which hinder diffusion and diminishes adsorption of large dye molecules. The best adsorption properties were found for TBC-0.5 material with the highest contribution of bentonite which is characterized by the highest share of mesoporosity (0.39) in comparison to microporosity (0.61). In the case of the three other samples TBC-1, TBC-2, and TBC-3, one could find the comparable representation of micro- (0.71–0.81) and mesopores (0.20–0.12), however, there were larger differences in specific surface area values. Thus, it results in the following differentiation of the adsorption values: TBC-3 > TBC-2 > TBC-1. The experimental data were described by Langmuir and Freundlich isotherms. The determined L and F parameters are summarized in Table 4. The high $R^2$ values (above 0.98) show that the Langmuir model describes well the experimental data for all adsorption systems. The $R^2$ in the range of 0.783–0.957 proved that the Freundlich model also well describes the experimental systems. The optimized adsorption capacities $a_m$ and $n$ parameters of the biochars containing bentonite in the structure were 3 to 5 times higher than for the material obtained by tannin pyrolysis. The highest $a_m$ value was obtained for the TBC-0.5 sample with the lowest micropore contribution, but the highest proportion of mesopores from the obtained materials, therefore, in the case of the carbonized material, the dye molecules can well penetrate pores. In addition, it was observed that in the case of carbonized material with the highest tannin content, the increase in MB adsorption was likely also affected by an increase in the specific surface area. The L and F parameters were compared to the literature data and shown in Table 5.

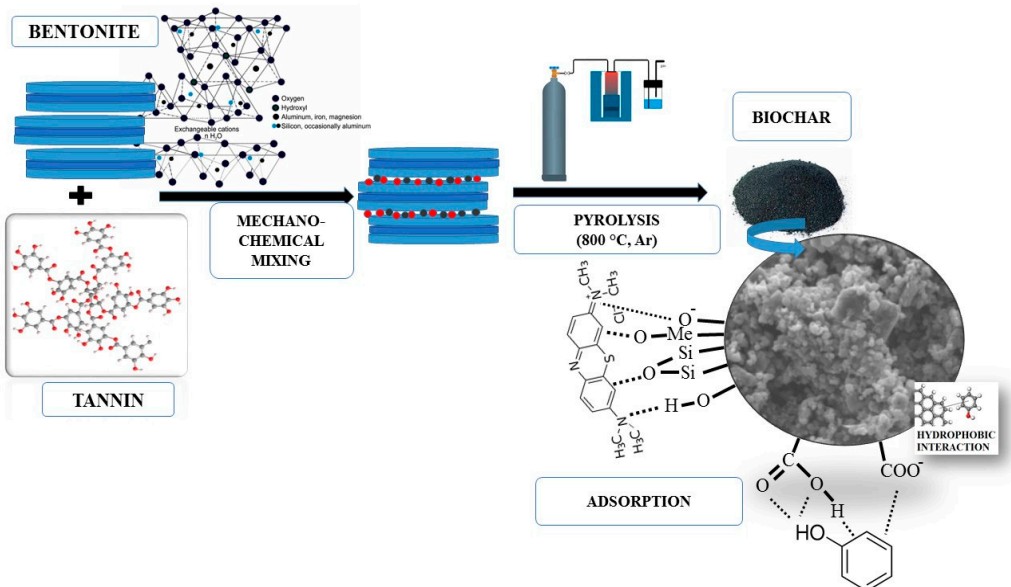

**Figure 6.** The scheme of synthesis of biochars from tannin and bentonite with possible mechanisms for methylene blue and phenol adsorption.

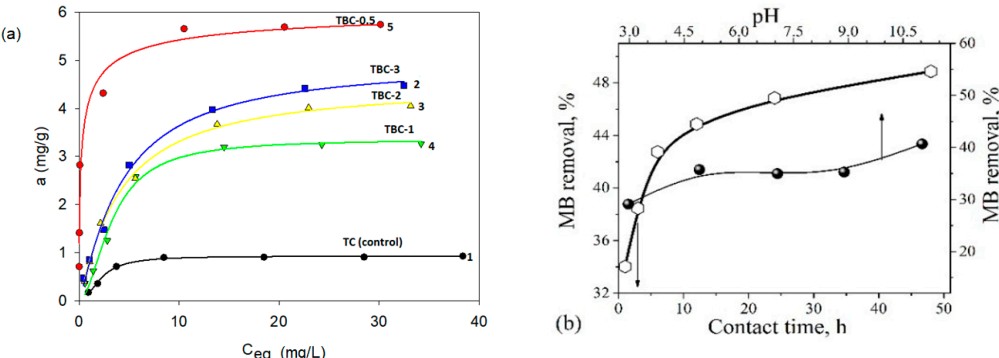

**Figure 7.** Comparison of adsorption isotherms of MB on the biochars: 1—TC (control), 2—TBC-3, 3—TBC-2, 4—TBC-1, 5—TBC-0.5 (**a**), and the effect of contact time and pH of the solution of MB on the removal by TBC-0.5 ($C_0$ = 20 mg/L, $m$ = 10 mg, $V_s$ = 5 mL) (**b**).

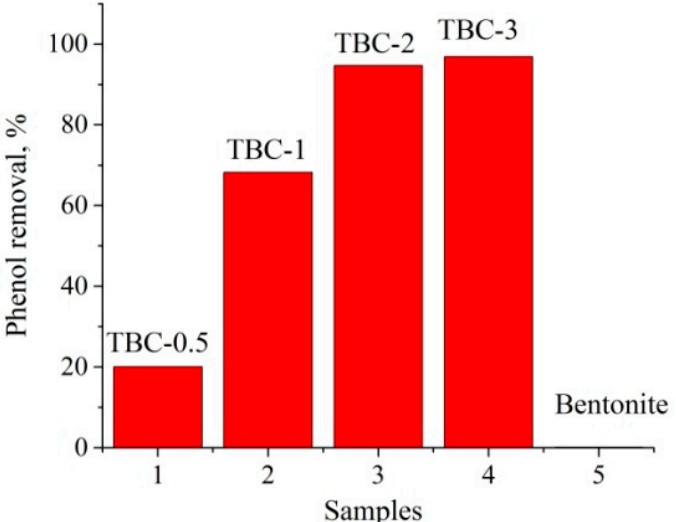

**Figure 8.** Phenol removal by TBC-0.5, TBC-1, TBC-3, and bentonite ($C_0$ = 20 mg/L, $m$ = 10 mg, $V_s$ = 5 mL).

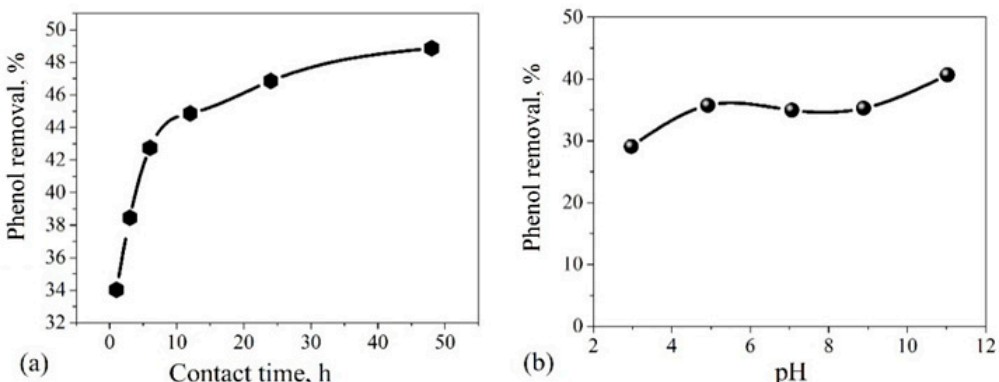

**Figure 9.** Effect of contact time (**a**) and pH (**b**) on the removal of phenol by TBC-0.5 ($C_0$ = 20 mg/L, $m$ = 10 mg, $V_s$ = 5mL).

**Table 4.** Isotherm parameters for adsorption of MB from water solutions.

| No. | Sample | Langmuir Isotherm | | |
|---|---|---|---|---|
| | | $R^2$ | $K_L$ (L/mg) | $a_m$ (mg/g) |
| 1 | TC (control) | 0.994 | 0.39 | 1.00 |
| 2 | TBC-0.5 | 0.999 | 3.61 | 5.78 |
| 3 | TBC-1 | 0.983 | 0.19 | 3.90 |
| 4 | TBC-2 | 0.996 | 0.21 | 4.74 |
| 5 | TBC-3 | 0.995 | 0.20 | 5.26 |
| | | Freundlich Isotherm | | |
| No. | Sample | $R^2$ | $K_F$ (mg/g) | $n$ |
| 1 | TC (control) | 0.783 | 0.28 | 2.50 |
| 2 | TBC-0.5 | 0.847 | 3.03 | 4.21 |
| 3 | TBC-1 | 0.907 | 0.60 | 1.77 |
| 4 | TBC-2 | 0.928 | 0.78 | 1.83 |
| 5 | TBC-3 | 0.957 | 0.87 | 1.85 |

Some differentiation in the sorption mechanisms of the cationic dyes should be taken into account: the hydrophobic interactions, electrostatic repulsion/attraction, oxidation-reduction reactions, or complex formation. Especially when taking into account the differentiation of acid–base properties of studied materials, the electrostatic interactions were of great importance. The MB was positively charged at the working pH and it interacted electrostatically with negatively charged and electron-rich functional groups of the adsorbents. To clarify this issue, additional measurements for phenol adsorption (Figure 8), as an organic compound with both hydrophobic and hydrophilic properties, were performed too. As can be seen in Figure 8, the mineral-rich biochars showed the opposite tendency toward phenol adsorption and the highest adsorption effectiveness was found for the samples with the highest carbon loading, while phenol poorly adsorbs on the raw bentonite. From the obtained data, it follows that the surface functional groups of the synthesized materials have a significant influence on the adsorption capacity if adsorbents have appropriate textural characteristics. In particular, in the case of methylene blue adsorption, the main centers involved in the interaction were the functional groups from clay: hydrogen bond between hydroxyl groups (Si-O-H) and nitrogen atoms of MB; π-π/n-π interaction between the aromatic structure of MB and carbons, as well as electron clouds of the Si-O-Si bridges; hydrogen bonding between hydroxyls (C-OH) and nitrogen of the dye molecule could also be attributed to it adsorption [42]. Thus, the mineral-enriched biochar (with higher bentonite content, sample TBC-0.5) shows a higher adsorption capacity for MB compared to the carbon-rich biochar (sample TBC-3). Schematic interpretation of MB interactions with biochar structures is shown in Figure 6. Phenol removal can take place due to the presence of carbon phase in the pyrolyzed materials: π–π dispersion interaction derived from phenol aromatic rings and carbon layers; physical adsorption based on π–π interaction; and via an electron donor-acceptor complex between surface C-C, C-O and C=O groups which can act as the electron donors and phenol aromatic rings (acceptors), that may occur simultaneously. Additionally, the C-OH group from phenol can form hydrogen bonds with O-containing carbon functionalities. Since phenol interacts with the carbon surface through the aromatic ring, the interactions will be weaker as carbon content decreases, which is observed in the case of the TBC-0.5 sample.

**Table 5.** Isotherm parameters for adsorption of MB from water solutions for selected adsorbents based on the literature data.

| No. | Sample | Langmuir Isotherm | | | |
| | | $R^2$ | $K_L$ (L/mg) | $a_m$ (mg/g) | Literature Source |
|---|---|---|---|---|---|
| 1 | EDTA modified bentonite | 0.998 | 3.45 | 160.00 | [43] |
| 2 | Modified Pumice Stone | 0.992 | 0.13 | 15.87 | [44] |
| 3 | Pumice | 0.975 | - | 0.02 | [45] |
| 4 | SAC | 0.991 | 0.14 | 126.60 | [46] |
| 5 | NR90CS10 blend | - | 0.94 | 1.12 | [47] |
| 6 | Natural clay | - | - | 4.98–50.00 | [48] |

| No. | Sample | Freundlich Isotherm | | | |
| | | $R^2$ | $K_F$(mg/g) | $n$ | Literature Source |
|---|---|---|---|---|---|
| 1 | EDTA modified bentonite | 0.774 | 84.8 | 3.61 | [43] |
| 2 | Modified Pumice Stone | 0.999 | 3.45 | 2.18 | [44] |
| 3 | Pumice | 0.986 | - | 1.47 | [45] |
| 4 | SAC | 0.987 | 41.7 | 4.76 | [46] |
| 5 | NR90CS10 blend | - | 0.47 | 2.32 | [47] |

## 4. Conclusions

In this work, textural, structural, morphological, and thermal characteristics of novel hybrid biochars prepared by mechanochemical activation and pyrolysis of natural origin constituents (tannin and bentonite), with an application as a sorbent system for methylene blue dye were described. Additionally, the phenol adsorption studies were performed for a better understanding of the sorption mechanism.

The XRD diffraction data confirm that the main crystalline phase of the studied materials corresponds to montmorillonite. An increase in tannin content in the reaction blends leads to an increase in the carbon content in the biochars. The Raman analysis reveals that the crystalline structure of bentonite undergoes significant changes during the tannin pyrolysis, with the simultaneous formation of a carbon phase with a disordered structure (D/G ratio is ~0.8), confirming that the carbon layers are mainly amorphous with many defects. The FT-IR analysis confirms the existence of characteristic stretching and deformation vibrations of the bentonite phase. Changes in peaks intensity and bands shifts are due to the destruction of the layered structure of bentonite during tannin carbonization. The SEM analysis shows that the carbon forms irregularly shaped particles of 1–1.5 μm sizes, and consists of amorphous carbon crystallites. With increasing the tannin content (a carbon source), the size of carbon globules does not practically change, but the bulk density of the biochars increases. The low-temperature nitrogen sorption studies indicate that materials are characterized by the BET specific surface area from 84 to 380 $m^2$/g. The changes in the textural characteristics of pyrolyzed materials are connected to the differences in the initial blend compositions. Potentiometric titration studies reveal an inert of TC (control) ($pH_{PZC}$ = 6.95), acidic of TBC-3 ($pH_{PZC}$ = 5.84), TBC-2 ($pH_{PZC}$ = 6.39), and TBC-1 ($pH_{PZC}$ = 6.41), and basic character of TBC-0.5 ($pH_{PZC}$ = 8.55), and bentonite and pyrolyzed bentonite ($pH_{PZC}$ = 7.90 and 8.42) surfaces. The thermal analysis of biocarbons shows that the main destruction of analyzed materials follows at the temperature range of 350–650°C. The carbon obtained only from tannin has the lowest $T_{max}$ = 550°C, however, for the materials with different carbon content, a slight differentiation of $T_{max}$ is observed. Generally, data obtained by XRD, Raman and FT-IR spectroscopies, potentiometric titration, SEM microscopy, and porosimetry are in good agreement. The presented studies confirm that bentonite affects the structurization of carbon, and the thermal stability of the materials is due to the Si–O–C bond formation in the sample and nanocarbon intercalation between the clay layers during the carbonization.

The adsorption data were examined by the Langmuir and Freundlich isotherms. The high $R^2$ values in the range of 0.983–0.999 show that the Langmuir model describes well

the experimental data for all adsorption systems. The R$^2$ in the range of 0.783–0.957 proved that the Freundlich model also well describes the experimental systems. The adsorption properties of obtained materials towards methylene blue are differentiated and correlations with their structural characteristics are found. Much better adsorption effectiveness is observed for the materials obtained with the addition of aluminosilicate with a maximum adsorption capacity of 5.78 mg/g compared to the carbon synthesized only from tannin ($a_m$ = 1.00). The obtained results indicate that the surface functional groups of the adsorbent are more critical for the adsorption of the studied organic substances than the surface area and porosity. They show differentiated affinity towards organic substances. The obtained materials are very good adsorbents for typical organics of hydrophobic properties which are poorly adsorbed on raw clay. The investigated materials can be used as an effective selective adsorbent for the removal of organic substances of differentiated properties from the aqueous solutions.

From a practical point of view, the obtained materials may be valuable adsorbents in water and wastewater purification technologies.

**Author Contributions:** Conceptualization, M.G. and V.B.; Material synthesis, M.G.; Investigation, M.G., A.B., D.S. and O.O.; Methodology, M.G., A.B., D.S., O.O., V.G. and A.D.-M.; Writing—original draft, M.G., A.B. and D.S.; Writing—review and editing, A.D.-M. and V.G. All authors have read and agreed to the published version of the manuscript.

**Funding:** This research received no external funding.

**Institutional Review Board Statement:** Not applicable.

**Informed Consent Statement:** Informed consent was obtained from all subjects involved in the study.

**Data Availability Statement:** The data and samples are available from the authors.

**Acknowledgments:** The authors are grateful for the National Academy of Science of Ukraine and the Polish Academy of Science, International Research Staff Exchange Scheme between NAS of Ukraine and PAS (2022–2024) for financial support of the project Pesticide removal from aqueous solutions using innovative composites.The authors thank Wojciech Franus from the Faculty of Civil Engineering and Architecture at the Lublin University of Technology for SEM analysis of the samples.

**Conflicts of Interest:** The authors declare no conflict of interest.

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
