# Peer review of "Development, Synthesis and Characterization of Tannin/Bentonite-Derived Biochar for Water and Wastewater Treatment from Methylene Blue"

_water, doi:10.3390/w14152407_

Round 1

Reviewer 1 Report

Introduction 

The author should rewrite the introduction in a better way than that, it looks like no relation between the paragraphs and there is more popular information, author should connect the paragraphs in a good way 

Materials and methods

2-1 – the author did not mention all the chemicals or instruments used in the work   

Only explain the structure and properties of MB which is not needed and all mentioned information about MB is popular.  my suggestion is omit the figure 1 and table 1.  

2.2. methods

From line 155-158

-          What the concentration of MB that used in this work,  and did the author use only one concentration of methylene blue?

-          It is mentioned that (7 days) this a period was taken to achieve absorbance equilibrium it is very long time according 

-          No schemes illustrate the preparation materials or absorbance mechanisms

-          I suggest using SEM microscopic photography to show the surface porosity of tannin/bentonite materials

-           

Line 166 (Result)

Write in subtitle

For example, 3- Result was subtitled as characterization and applications

Line 281 the applications of materials in the study is the absorbance of  MB it needs more explanation

-          The concentration effect of MB in the absorbance process and what is the concentration of MB which used in the study

-          The author did not mention the effect of pH change in the absorbance process  

-          Only Langmuir isotherm was used in this study, what about the other absorbance isotherms?

-          No kinetic studies in this work

No 

Author Response

[Water] Manuscript ID: water-1813640

Authors: Mariia Galaburda, Alicja Bosacka, Dariusz Sternik, Viktor Bogatyrov, Dariusz Sternik, OlenaOranska, Vladimir Gun’ko and Anna DeryÅ‚o-Marczewska

Title: Development, Synthesis And Characterization of Tannin/Bentonite-Derived Biochar for Water and Wastewater treatment from Methylene Blue

Response to Reviewer 1 Comments

Introduction 

The author should rewrite the introduction in a better way than that, it looks like no relation between the paragraphs and there is more popular information, author should connect the paragraphs in a good way 

Thank you for your comment. The introduction was rewritten. We have tried to connect the paragraphs in a better way.

Materials and methods

2-1 – the author did not mention all the chemicals or instruments used in the work   

2.2. methods

From line 155-158

Thank you for your comment. The information about instruments and chemicals have been corrected.

Only explain the structure and properties of MB which is not needed and all mentioned information about MB is popular.  my suggestion is omit the figure 1 and table 1.

Thank you for your comment. Figure 1 and Table 1 were omitted. Some information about methylene blue was moved to the Introduction. We hope this information is sufficient.

-          What the concentration of MB that used in this work,  and did the author use only one concentration of methylene blue?

Thank you for your comment. The concentration was added to the experiment description and itwas in the range of 1.25-40 mg/L.

-          It is mentioned that (7 days) this a period was taken to achieve absorbance equilibrium it is very long time according 

Thank you for your comment. The experiment was performed for 48 hours (2 days). It was a mistake. Thank you very much for paying attention.

-          No schemes illustrate the preparation materials or absorbance mechanisms

Thank you for your comment. The scheme was added based on data received from the methylene blue adsorption and additionally phenol adsorption studies.

-          I suggest using SEM microscopic photography to show the surface porosity of tannin/bentonite materials

Thank you for your comment. The SEM micrographs were added.

Line 166 (Result)

Write in subtitle

For example, 3- Result was subtitled as characterization and applications

Thank you for your comment. The subtitles in this section were added.

Line 281 the applications of materials in the study is the absorbance of  MB it needs more explanation.

Thank you for your comment. We have tried to moreexplain the application of the materials in the adsorption of methylene blue than the previously.

-          The concentration effect of MB in the absorbance process and what is the concentration of MB used in the study

Thank you for your comment.  The concentration was added in the experiment description and itis in the range of 1.25-40 mg/L.

-          The author did not mention the effect of pH change in the absorbance process 

Thank you for your comment. The pH effect and also contact time of MB onto the TBC-0.5 sample were additionally studied.

-          Only Langmuir isotherm was used in this study, what about the other absorbance isotherms?

Thank you for your comment.The Freundlich isotherm was additionally applied.

-          No kinetic studies in this work

Thank you for your comment. It is true. Only equilibrium adsorption studies were carried out.Due to time restrictions and small amounts of materials we could not perform the kinetic experiments. However, the contact time and pH effect studies were additionally performedfor MB and phenol adsorption was also studied.

Reviewer 2 Report

The presented manuscript includes the study of the development, synthesis, and characterization of tannin/bentonite-derived biochar for water and wastewater treatment from methylene blue.

The results of the work are presented on a good level and well written, but, some general corrections and weak points should be mentioned.

1. Title. What kind of water can include methylene blue? If authors will find some suitable, is it really a common problem?

2. Abstract. “Green chemistry technology” should be proved. Mention absence of solvent means nothing, including energy-consuming ball milling and pyrolysis. Why more expensive argon was used instead of nitrogen?

3. Abstract. “nanostructured” should be proved.

4. Strange phrase “the economical and ecological tannin-bentonite (nano)materials obtained by mechanochemical activation and pyrolysis”. Please clarify the idea of the sentence.

5. The X-axis in fig. 3 should be reversed (started from 4000 on the left and lowest values on the right side of the axis).

6. Why the temperature of 800 C was set for pyrolysis. Based on fig. 4 it would be enough to set 650 C.

7. Fig. 5 is incorrectly classified as a Type IV isotherm. I mean that type IV is split into 2 different subtypes. Please have a look at the IUPAC recommendations for describing isotherms (Pure Appl. Chem. 2015, 87, 1051 DOI:10.1515/pac-2014-1117) and revise the classification. Please, also correct this in all parts of the text body (“with IV type of isotherm and H4 hysteresis loop (IUPAC classification)”).

8. “To sum up, the obtained low-cost, safe, and environmental friendly tannin(carbon)/bentonite biochars show differentiated selectivity towards methylene blue and much better adsorption effectiveness in comparison to the carbon obtained by tannin pyrolysis.” – please prove i) “low-cost”, ii) “safe”, iii) “selectivity towards methylene blue”.

9. “In this work, porous, structural, and thermal characteristics of novel ecological and economical hybrid…” – from the point of view of ecology and economy, these words “novel ecological and economical”  in this sentence do not make any sense. 

10. What about regeneration and final utilization of spent adsorbent?

11. Please, also, compare your results with published ones by a couple of parameters (usually it is presented at the end of the discussion section in the form of a table).

Your paper is pretty comprehensive. Please, do not make it worth with not proven statements like “ecological”, “economic”, “nanostructured” etc. This is very common in post-soviet countries to give characteristics without any proof and looks at minimum very strange.

Author Response

[Water] Manuscript ID: water-1813640

Authors: Mariia Galaburda, Alicja Bosacka, Dariusz Sternik, Viktor Bogatyrov, Dariusz Sternik, OlenaOranska, Vladimir Gun’ko and Anna DeryÅ‚o-Marczewska

Title: Development, Synthesis And Characterization of Tannin/Bentonite-Derived Biochar for Water and Wastewater treatment from Methylene Blue

Response to Reviewer 2 Comments

The presented manuscript includes the study of the development, synthesis, and characterization of tannin/bentonite-derived biochar for water and wastewater treatment from methylene blue.

The results of the work are presented on a good level and well written, but, some general corrections and weak points should be mentioned.

  1. What kind of water can include methylene blue? If authors will find some suitable, is it really a common problem?

Thank you for your comment. As was written in the introduction methylene blue is an environmentally dangerous and toxic substance for human health. Of course, there are attempts to use other safer dyes, but numerous publications prove that it is still a commonly used dye for fabrics on blue.

Generally, methylene blue dye can be found in industrial waters. However, contaminants may find their way into rivers and further into the seas due to leaks in installations, etc.

  1. “Green chemistry technology” should be proved. Mention absence of solvent means nothing, including energy-consuming ball milling and pyrolysis. Why more expensive argon was used instead of nitrogen?

Thank you for your comment. Considering the energy consumption aspect, we could agree, however, not using often toxic organic solvents in wet synthesis, still is an advantage. All statements such as green chemistry, economical, ecological have been removed from the text.

In terms of gas used to the synthesis, the use of argon was depended upon the availability of the laboratory. In our work with air-sensitive chemistry, (investigation of transformation of organic groups of organometallics) there can be some advantage to argon due to its higher density [argon (1.714 g/L), nitrogen (1.251 g/L)], meaning it tends to blanket the reaction and argon doesn't escape so quickly if you need to quickly open the flask to put a reagent in. For inorganic chemistry, there are situations where nitrogen will react with certain chemicals, and in these cases, argon must be used. Nitrogen from cylinders can contain traces of oxygen and water vapor, which may be important in our application.

Given the Praxir argon has 1 ppm less oxygen than the Praxair Nitrogen.

  1. “nanostructured” should be proved.

Thank you for your comment.The formation of nanostructures was confirmed using a more detailed analysis of XRD, Raman spectroscopy and SEM analysis. However, the word nanostructure was tried to be omitted in the text.

  1. Strange phrase “the economical and ecological tannin-bentonite (nano)materials obtained by mechanochemical activation and pyrolysis”. Please clarify the idea of the sentence.

Thank you for your comment. Based on comment 2, we rearrange all similar sentences in the manuscript.

  1. The X-axis in fig. 3 should be reversed (started from 4000 on the left and lowest values on the right side of the axis).

Thank you for your comment. The X-axis was reversed and starts from 4000 cm−1.

  1. Why the temperature of 800 ºC was set for pyrolysis. Based on fig. 4 it would be enough to set 650 °

Thank you for your comment. Based on the results presented in Figure 4, the destruction of carbonized composites on the air was described. Pyrolysis temperature had a strong influence on biochar physicochemical properties. It was reported that higher pyrolysis temperature (i.e., more than 700 °C) C=C bond breakages take place due to the availability of adequate energy. Therefore, at higher temperature, due to extensive carbonization, the formation of graphite-like structures of the biochar can occur, which shows less intense peaks on the surface and high percentages of carbon content. The structure of biochar appears to have more organized carbon layers. It increases the specific surface area and pore volume.

  1. 5 is incorrectly classified as a Type IV isotherm. I mean that type IV is split into 2 different subtypes. Please have a look at the IUPAC recommendations for describing isotherms (Pure Appl. Chem. 2015, 87, 1051 DOI:10.1515/pac-2014-1117) and revise the classification. Please, also correct this in all parts of the text body (“with IV type of isotherm and H4 hysteresis loop (IUPAC classification)”).

Thank you for your comment.The classification of isotherms was updated.

  1. “To sum up, the obtained low-cost, safe, and environmental friendly tannin(carbon)/bentonite biochars show differentiated selectivity towards methylene blue and much better adsorption effectiveness in comparison to the carbon obtained by tannin pyrolysis.” – please prove i) “low-cost”, ii) “safe”, iii) “selectivity towards methylene blue”.

Thank you for your comment. The sentence was removed.

  1. “In this work, porous, structural, and thermal characteristics of novel ecological and economical hybrid…” – from the point of view of ecology and economy, these words “novel ecological and economical”  in this sentence do not make any sense.

Thank you for your comment. Based on the comments 2 and 4, we rearrange all similar sentences in the manuscript.

  1. What about regeneration and final utilization of spent adsorbent?

Thank you for your comment.Due to time restrictions and small amounts of materials we could not perform the regeneration experiments.

Please, also, compare your results with published ones by a couple of parameters (usually it is presented at the end of the discussion section in the form of a table).

Thank you for your comment. Some literature data in the form of Table were added.

  1. Your paper is pretty comprehensive. Please, do not make it worth with not proven statements like “ecological”, “economic”, “nanostructured” etc. This is very common in post-soviet countries to give characteristics without any proof and looks at minimum very strange.

Thank you for your comment. Thank you for pointing the way. In the future, we will try to avoid such phrases.

Reviewer 3 Report

My comments are list in the attached file

Author Response

[Water] Manuscript ID: water-1813640

Authors: Mariia Galaburda, Alicja Bosacka, Dariusz Sternik, Viktor Bogatyrov, Dariusz Sternik, Olena Oranska, Vladimir Gun’ko and Anna DeryÅ‚o-Marczewska

Title: Development, Synthesis And Characterization of Tannin/Bentonite-Derived Biochar for Water and Wastewater treatment from Methylene Blue

Response to Reviewer 3 Comments

My comments are list below:

  1. Add some quantitative results in the Abstract and Conclusion sections

Thank you for your comment. The qualitative results were added in the Abstract and Conclusion.

  1. Please add the novelty of the study

Thank you for your comment.The novelty aspects were added.

  1. L108-109: The sentence can be rewrite as follow: ‘’ Two-stage methods of synthesis were used: (i) mechanochemical activation of a mixture and (ii) pyrolysis’’ L109: Please remove ‘’(i)’’ L111: Please remove ‘’(ii)’’

Thank you for your comment. The sentence was rewritten and numbers were removed.

  1. L113: Please rewrite: ‘’The heating rate was 10 °C/min up to 800 °Ð¡ and the sample…..’’

Thank you for your comment. The sentence was rewritten correctly.

  1. All the figures provided are of low resolution.

Thank you for your comment. The Figures were changed on those with higher resolution.

  1. L156: Please indicate the values of MB concentrations used for the adsorption study

Thank you for your comment. The concentration of MB was added in the experiment description and was itin the range of 1.25 to 40 mg/g.

  1. L185-186: ‘’The spectra of biochar are characterized by two distinctive bands, a G (at ~1600 cm-1), and a D (at ~1359 cm–1) band [31].’’ From Figure 2b the peak corresponding to D band is noted at 1350 cm-1. Please check

Thank you for your comment. The value was corrected at 1350 cm−1.

  1. Please include in the Figure 2b the results for bentonite pyrolyzed, and bentonite

Thank you for your comment. The results forbentonite pyrolyzed and bentonite were included in Figure 2b.

  1. L199: Add a space between ‘’tannin’’ and ‘’ (carbon)

Thank you for your comment. The space was added in this place.

  1. Please rewrite the sentence: ‘’BET surface area is equal 86, 128, 194, and 230 m²/g for TBC-0.5, TBC-1, TBC-2, 249 TBC-3, and 380, 86 m²/g for TC (control) and bentonite, respectively.’’

Thank you for your comment. The sentence was rewritten.

  1. L272: From the Table 4 it can be seen that the value of pHPZCof bentonite is 7.9. Please check

Thank you for your comment. The value was corrected.

  1. Add the equation used for adsorption capacity, mg/g

Thank you for your comment. The given equation was added.

  1. I suggest to add the results for TG/DTG analysis and adsorption isotherms of MB onto bentonite material taking into account that XRD, Raman spectroscopy, FTIR, and Nitrogen adsorption/desorption isotherms are presented in the manuscript

Thank you for your comment.It was done.

  1. Please add the value of pH used in the adsorption study

Thank you for your comment. The pH value was added in experiment description.

  1. Please add at least another isotherm model for comparison (for example Freundlich)

Thank you for your comment. The Freundlich isotherm was added.

  1. For TBC-0.5 material add the study regarding the effect of contact time in order to establish the contact time necessary to attain the equilibrium. The study for the effect of initial concentration was performed at a contact time of 7 days. Also, please add a kinetic study

Thank you for your comment. The experiment was performed for 48 hours (2 days). It was a mistake. Thank you very much for paying attention. Only equilibrium adsorption studies were carried out. Due to time restrictions and small amounts of materials we could not perform the kinetic experiments.However, the recovery, contact time and pH effect studies were additionally performed for MB and phenol adsorption was also studied.

  1. Add a comparison with the adsorption capacities, mg/g of the proposed materials with other adsorbents used for MB removal already presented in the literature

Thank you for your comment. Some literature data in the form of a Table were added.

  1. Add a regeneration study of the TBC-0.5 adsorbent.

Thank you for your comment. Due to time restrictions and small amounts of materials we could not perform the regeneration experiments.

  1. Pleasecheck the Reference 14

Thank you for your comment. The reference was checked and corrected.

  1. The writing of bibliographic notes differs

Thank you for your comment. The bibliography was corrected in the manuscript.

Round 2

Reviewer 1 Report

The paper has been improved more ,

but you should concern to introduction paragraphs

and mention the adsorption process was applied with Langmuir isotherm or Freundlich isotherm

the  conclusions is more long , it should be summarized and focused on final results 

Author Response

Thank you for your comments. We tried to correct the work according to the comments. In the abstract, introduction, and conclusions it was mentioned that the the adsorption data were analyzed applying Langmuir and Freundlich isotherms. The text was checked in terms of the English language. The conclusions were shortened. In this section, the most important information summarizing the results was included.

Reviewer 2 Report

thank you for corrections

Author Response

Thank you for your comment.

Reviewer 3 Report

The manuscript has been improved as suggested.

Author Response

Thank you for your comment.